# A Recognition Method of Ewe Estrus Crawling Behavior Based on Multi-Target Detection Layer Neural Network

**DOI:** 10.3390/ani13030413

**Published:** 2023-01-26

**Authors:** Longhui Yu, Jianjun Guo, Yuhai Pu, Honglei Cen, Jingbin Li, Shuangyin Liu, Jing Nie, Jianbing Ge, Shuo Yang, Hangxing Zhao, Yalei Xu, Jianglin Wu, Kang Wang

**Affiliations:** 1College of Mechanical and Electrical Engineering, Shihezi University, Shihezi 832003, China; 2Xinjiang Production and Construction Corps Key Laboratory of Modern Agricultural Machinery, Shihezi 832003, China; 3Industrial Technology Research Institute of Xinjiang Production and Construction Corps, Shihezi 832000, China; 4College of Information Science and Technology, Zhongkai University of Agriculture and Engineering, Guangzhou 510225, China

**Keywords:** behavior recognition, deep learning, ewe estrus, target detection, YOLO v3

## Abstract

**Simple Summary:**

The timely and accurate detection of ewe estrus behavior in precision animal husbandry is an important research topic. The timely detection of estrus ewes in mutton sheep breeding can not only protect the welfare of ewes themselves, but also improve the interests of breeding enterprises. With the continuous increase in the scale of mutton sheep breeding and the gradual intensification of breeding methods, the traditional manual detection methods require high labor intensity, and the contact sensor detection methods will cause stress reaction problems in ewes. In recent years, the rapid development of deep learning technology has brought on new possibilities. We propose a method for the recognition ewe estrus based on a multi-target detection layer neural network. The results show that the method can meet the requirements of timely and accurate detection of ewe estrus behavior in large-scale mutton sheep breeding.

**Abstract:**

There are some problems with estrus detection in ewes in large-scale meat sheep farming: mainly, the manual detection method is labor-intensive and the contact sensor detection method causes stress reactions in ewes. To solve the abovementioned problems, we proposed a multi-objective detection layer neural network-based method for ewe estrus crawling behavior recognition. The approach we proposed has four main parts. Firstly, to address the problem of mismatch between our constructed ewe estrus dataset and the YOLO v3 anchor box size, we propose to obtain a new anchor box size by clustering the ewe estrus dataset using the K-means++ algorithm. Secondly, to address the problem of low model recognition precision caused by small imaging of distant ewes in the dataset, we added a 104 × 104 target detection layer, making the total target detection layer reach four layers, strengthening the model’s ability to learn shallow information and improving the model’s ability to detect small targets. Then, we added residual units to the residual structure of the model, so that the deep feature information of the model is not easily lost and further fused with the shallow feature information to speed up the training of the model. Finally, we maintain the aspect ratio of the images in the data-loading module of the model to reduce the distortion of the image information and increase the precision of the model. The experimental results show that our proposed model has 98.56% recognition precision, while recall was 98.04%, F1 value was 98%, mAP was 99.78%, FPS was 41 f/s, and model size was 276 M, which can meet the accurate and real-time recognition of ewe estrus behavior in large-scale meat sheep farming.

## 1. Introduction

At present, the feeding mode of mutton sheep mainly based on small-scale scattered breeding of farmers is the main factor restricting the growth of mutton production. The large-scale and intensive breeding mode proposed by precision animal husbandry is an important measure and guarantee for increasing mutton production. Artificial insemination can be carried out at the most appropriate time through the timely and accurate detection of ewe estrus in large-scale and intensive farming, thereby reducing the interval between birth and fertilization costs [1,2,3]. Therefore, the timely and accurate detection of estrus in precision animal husbandry is a key technology [4,5]. The traditional method is to observe the activity and physiological status of estrus ewes manually [6,7,8]. The traditional manual observation method is not only time-consuming and labor-intensive, but also vulnerable to subjective experience. Domestic and international research methods on animal estrus behavior recognition mainly include sensor-based research methods and computer vision-based research methods.

Researchers have proposed a sensor-based technique for recognizing animal estrus behavior based on the physiological changes and behavioral characteristics of animals during estrus; along with the properties of sensors [9,10,11,12,13,14,15,16,17,18,19], movement [11,12,13,14,15,16,17], temperature [18], and sound [19], these are the most common approaches. The sensor-based research approach may easily gather data on animals in estrus. However, the sensors are often contact-type, causing the animals to create stress reactions, entailing high technical requirements for breeding personnel and considerable device costs.

In recent years, with the proposed video-image-processing algorithms in the field of computer vision, researchers have used a method based on video image processing to identify animal estrus behavior [20,21,22,23]. The video image processing-based method of animal estrus behavior recognition reduces labor costs and improves animal welfare, but the method makes it difficult to extract features manually. With the increase in computing power and the booming development of deep learning technology, researchers have carried out research in various aspects based on deep learning technology [24,25,26,27,28,29,30,31,32,33,34,35,36,37]. Gan et al. proposed using optical flow and feature estimation based on cascaded convolutional neural network to detect the nursing area of sow, and using RGB images and corresponding optical flow images of the nursing area to detect the nursing behavior of sow [24]. Song et al. improved the YOLO v3 model via re-clustering anchor frame and channel shearing, and realized the face recognition of sheep [25]. Wang et al. improved faster R-CNN through key frame extraction, foreground segmentation and region recommendation. The experimental results show that the accuracy of the improved method was 92.57%, and the detection speed is twice faster [26]. Duan et al. used long short-term memory (LSTM) networks to classify the sound of short-term feeding behavior of sheep, including biting, chewing, bolus regurgitation, and rumination chewing [27]. Chen et al. used Xception and an LSTM model to extract the spatial and temporal characteristics of the target, and then used maximum entropy segmentation and an HSV color space transformation algorithm to identify pigs and feed time [28]. Wu et al. used YOLO v3 model to extract the leg area coordinates of dairy cows in video frames, calculate the relative step length of the front and rear legs of dairy cows, and construct the relative step length feature vector, and then used LSTM model to classify lameness [29]. Lu et al. proposed a maturity calculation method of winter jujube combining the YOLO v3 model and artificial features, and the accuracy of maturity classification was 97.28% [30]. He et al. proposed an analysis method of cow estrus in endoscopic images based on CNN [31]. According to the characteristics of the sheep face dataset, Noor et al. used transfer learning method to train a CNN model and realized the classification of normal (no pain) and abnormal (pain) sheep images [32].

In summary, we can see that deep learning technology can not only automatically obtain multi-dimensional and multi-level feature information of the target in the image, but also effectively solve the problem of manually extracting the target feature vector, which makes deep learning technology have a broad application prospect in the field of precision animal husbandry. Among the deep learning techniques, the YOLO v3 model is a representative of target detection methods. YOLO v3 has the advantages of fast recognition speed, high recognition accuracy, and strong model resistance to interference, and has the potential to recognize ewe estrus crawling behavior in complex environments. Therefore, we chose YOLO v3 for the detection of ewe estrus.

To address the shortcomings of manual detection methods and sensor detection methods in the detection of ewe estrus, we proposed a multi-objective detection layer neural network-based method for ewe estrus crawling behavior recognition. Firstly, we decomposed the ewe estrus videos obtained from large-scale meat sheep farms into pictures using the video keyframe decomposition technique. Secondly, we used LabelImg to manually annotate the images containing ewe estrus crawling behavior to obtain the sample dataset. Then, we improved the model based on the dataset and YOLO v3 model structure characteristics. We mainly improved the model in terms of target detection layer, residual structural unit, anchor box size, and image resolution-keeping aspect ratio. Finally, we train, validate, and test the improved model using the PyTorch deep learning framework, the PyCharm development tool, and a high-performance computing platform.

Briefly, the main contributions of this work are summarized as follows:

(1) We established an ewe estrus dataset, along with the thorough analysis and findings on our dataset.

(2) We propose to use the YOLO v3 model to detect the estrus crawling behavior of ewes, and improve the YOLO v3 model.

(3) The advantages and disadvantages of our proposed model are compared with other models.

The rest of this paper is organized as follows. In Section 2, we describe in detail how to produce the sample dataset and make improvements to the network model. In Section 3, we present the experimental results of the model and discuss the results in detail. Finally, Section 4 concludes the paper.

## 2. Materials and Methods

### 2.1. Materials Preparation

#### 2.1.1. Data Sources

The video of the estrus crawling behavior of ewes in this study was collected from the Xinao animal husbandry and mutton sheep breeding base in Lanzhou Bay Town, Manas County, Xinjiang Uygur Autonomous Region. Through a field investigation, we found that the estrus crawling behavior of ewes mainly occurred in the shaded area; therefore, this study chose to be in the shaded area for video capture. The shaded area is 33.75 m long and 13 m wide. Two surveillance cameras with a resolution of 4 million pixels (Hikvision DS-2CD2346FWD-IS dome network cameras) (Hangzhou Hikvision Digital Technology Co., Hangzhou, China) are installed in the shaded area, and the camera installation height is 2.7 m.

The installed camera and field of view are shown in Figure 1. The video of estrus behavior of ewes collected the activity records of 5 adult rams and 80 adult ewes. The video image collection time was from May 2021 to November 2021. The video resolution is 2560 pixels × 1440 pixels, and the frame rate is 25 f/s.

#### 2.1.2. Image Preprocessing

When we watched the video image of ewe estrus, we found that the video image taken by camera 2 contained more unrelated areas. In the process of ewe estrus detection, these more unrelated regions will increase the amount of calculations required and error recognition rate. On the one hand, the video shooting position and camera rotation angle are fixed; on the other hand, the ewes’ activity range is within the shaded area; therefore, there is no ewe estrus crawling behavior occurring outside the shade area. In summary, in order to reduce the computation and detect interference, we removed the redundant irrelevant regions in the video screen using a masking method, as shown in Figure 2.

#### 2.1.3. Dataset Construction

After watching the ewe activity monitoring video, the breeding staff selected 300 video clips with estrus climbing behavior from the video, with each clip ranging from 4 to 20 s in length. We used Python script to decompose video key frames of video clips, and take one frame every three frames to obtain 3640 original images. In these original images, we found that each image contains not only the crawling behavior of ewes in estrus, but also other behaviors of ewes not in estrus, such as feeding, standing, walking, lying down, etc.

In order to expand the sample size of the dataset and improve the generalization ability of the model, we performed data enhancement processing on the dataset. Firstly, in view of the uncertainty of the location of ewe’s climbing behavior in the shaded area, we performed horizontal flipping, horizontal moving, and vertical moving of the samples in the dataset. Then, according to the complex and variable illumination characteristics of the shaded area, brightness enhancement and brightness reduction were carried out. Data enhancement processing is shown in Figure 3. After the abovementioned data enhancement processing, we finally obtained 5600 sample images of the dataset.

#### 2.1.4. Dataset Label Making

Dataset label production is the first step before model training. Whether the production of dataset labels is standardized will directly affect the model training, verification, and test results. When labeling, the label box should only select the entire climbing behavior as much as possible. If the label box is too big, it will include too much noise, such as dust and backdrop, reducing the model identification’s precision.

We used the open-source labeling tool LabelImg to label the dataset and generate the corresponding xml file manually. After randomly dividing the obtained 5600 images according to the ratio of training, validation, and testing of 7:2:1 [38], there were 3920 images in the training set, 1120 images in the validation set, and 560 images in the test set. The dataset labels were created as shown in Figure 4.

### 2.2. Construction of an Estrus Detection Model for Ewes

#### 2.2.1. YOLO V3 Model

The YOLO v3 model modifies the YOLO v2 model presented by Redmon [39] in 2018. The model showed significant improvement in recognition speed and accuracy. YOLO v3 employs Darknet-53 as the model feature extraction network and the YOLO layer as the model target detection layer. Figure 5a illustrates the model structure.

YOLO v3 produces feature layers of three sizes from input images for feature extraction. If the input image size is 416 × 416 × 3, the feature layer sizes are 52 × 52, 26 × 26, and 13 × 13, respectively. The prediction target mechanism of multi-scale feature fusion was added in the target detection layer. Three feature fusion techniques were employed to gather the target’s location and semantic information; the detection is then performed on three distinct object detection layers. The picture was divided into grids of various sizes. Each grid was allocated three anchor boxes of varying sizes for bounding box prediction. The bounding box of the retrieved object has five predictions: x, y, w, h, and c. Where x, y is the target detection frame’s center coordinates, c is the confidence level, w, h is the detection frame’s width and height. Finally, the output tensors of the three target detection layers were merged, and the output is as shown in Equation (1):(1)$$\begin{array}{l} N\times \left(3\times \left(13\times 13+26\times 26+52\times 52\right)\right)\times \left(5+M\right)\\ =N\times 10,647\times \left(5+M\right) \end{array}$$

According to YOLO v3, target detection is performed on the COCO dataset, and the number of categories in the dataset is 80. The output is shown in Figure 5b.

#### 2.2.2. Construction of a Multi-Target Detection Layer Network Model

Because the camera position was fixed, we found that the ewe imaging occupies a smaller percentage in the video pictures of ewe estrus crawling behavior when the ewe is far away from the camera. The ewe target with a smaller imaging share in the picture will cause part of the relative position information to be lost after multi-layer convolution calculation, which affects the detection accuracy of the model for ewe estrus crawling behavior recognition.

When we directly used the original YOLO v3 model to detect distant ewe targets in estrus, the detection results did not reach a satisfactory level. Therefore, we need to improve the YOLO v3 model to improve the precision of the model for detecting distant ewes in video pictures.

Based on the three target detection layers of the original YOLO v3 model, we added a target detection layer with a size of 104 × 104. Our improved model has four target detection layers, where the 104 × 104 target detection layer is mainly responsible for detecting small targets in video pictures.

First, we obtained a 104 × 104 feature layer by convolving and upsampling the 52 × 52 feature layer after the operation. Then, we spliced the obtained 104 × 104 feature layer with the 104 × 104 feature layer in the feature extraction network using a concat function, and thus obtain the 104 × 104 target detection layer.

Our improved method helps the model extract more features from the shallow network and improves the model’s ability to detect distant estrus ewes in images. At the same time, there was some improvement in the wrong detection, missed detection and low confidence level of ewe estrus behavior identification. The model improvement locations are shown as Improvement 1 in Figure 6a.

The improved model has four target detection layers. Finally, the output tensors of the four target detection layers should be merged. For the input image, the output of the model is as shown in Equation (2):(2)$$\begin{array}{l} N\times \left(3\times \left(13\times 13+26\times 26+52\times 52+104\times 104\right)\right)\times \left(5+M\right)\\ =N\times 43,095\times \left(5+M\right) \end{array}$$

In Equations (1) and (2), N is the number of input images and M denotes the total number of categories in the dataset. The number of dataset categories in this paper is 1, so M is 1. The improved model output is shown in Figure 6b.

#### 2.2.3. Residual Structural Unit Optimization

The number of residual units in the residual structure of the original YOLO v3 model is 1, 2, 8, 8, and 4, respectively.

Since we added a 104 × 104 target detection layer, the improved model may experienced a loss of shallow feature information during the forward propagation training process, which in turn leads to a decrease in the fitting ability of the model. At the same time, the model may suffer from gradient dispersion and gradient explosion during the backpropagation training process, resulting in the inability to transfer deep feature information to shallow layers, which in turn leads to poor detection of the model.

Therefore, we add two residual unit to the last residual structure in the model, which is used to help the model extract more information about the deeper features of ewe estrus climbing behavior, thus increasing the model’s detection precision. The number of residual units in the residual structure of our improved model is 1, 2, 8, 8, and 6, respectively.

The optimized location of the residual structure unit is shown in Figure 6a in Improvement 2.

#### 2.2.4. Picture Resolution Maintains Aspect Ratio

The resolution size of the dataset image we created is 2560 pixels × 1440 pixels, and the data-loading module of the model will adjust the resolution of the input image to 416 pixels × 416 pixels. Therefore, the resolution of the image will be adjusted during the training, verification, and testing of the model.

If the resolution of the image is directly adjusted, it will not only cause serious distortion due to the stretching of the image, but also lead the model to extract the wrong features.

In order to reduce the adverse effect of picture distortion on the model and enable the model to extract effective features, we redefine the adjusted picture resolution function in the data-loading module and adopt the method of maintaining the horizontal and vertical ratio of the picture, as shown in Equations (3)–(6).
(3)$$w_{3}{=w}_{2}\times {\left(\frac{w_{1}}{w_{2}},\frac{h_{1}}{h_{2}}\right)}_{\min}$$
(4)$$h_{3}{=h}_{2}\times {\left(\frac{w_{1}}{w_{2}},\frac{h_{1}}{h_{2}}\right)}_{\min}$$
(5)$$d_{w}=\left(w_{1}{-w}_{3}\right)//2$$
(6)$$d_{h}=\left(h_{1}{-h}_{3}\right)//2$$

In Equations (3)–(6), w_1_ is the width of the image resolution required by the model, h_1_ is the height of the image resolution required by the model, w_2_ is the width of the image resolution of the dataset, h_2_ is the height of the image resolution of the dataset, w_3_ is the width of the picture adjusted to maintain the aspect ratio, h_3_ is the height of the picture adjusted to maintain the aspect ratio, d_w_ is the size of the pixels that need to be filled on the periphery of the image’s width, d_h_ is the size of the pixels that need to be filled on the periphery of the image’s height. When we carry out the picture fill, the fill pixel size is gray RGB (128, 128, 128). The dataset picture is shown in Figure 7 after adjusting with aspect ratio.

### 2.3. Re-Clustering of Anchor Boxes

The YOLO v3 model obtains nine anchor boxes in the COCO dataset using the K-means algorithm. Anchor boxes are a set of values with fixed width and height values. 

Among the anchor boxes, the three sizes of 10 × 13, 16 × 30 and 33 × 23 correspond to the 52 × 52 target detection layer and are responsible for detecting small targets. Among them, the three sizes of anchor boxes, 30 × 61, 62 × 45 and 59 × 119, correspond to the 26 × 26 target detection layer and are responsible for detecting medium-sized targets. Among them, the three sizes of anchor boxes, 116 × 90, 156 × 198 and 373 × 326, correspond to the 13 × 13 target detection layer and are responsible for detecting large targets.

During model training, validation and detection, three anchor boxes are assigned to each target detection layer. Each anchor box is computed in turn with the intersection and merge ratio (intersection over union, IOU) with the sample box containing the target center. Back propagation through the loss function adjusts the model parameters to continuously bring the prediction box close to the true box. Therefore, the size of the anchor box will directly affect the accuracy of the model detection. Due to the target size in the COCO dataset not matching the target size in this dataset, we need to re-cluster the anchor box size to improve the precision of model detection.

The K-means algorithm is an unsupervised learning method that uses distance as a similarity indicator. First, the width and height of the anchor boxes are selected as the clustering objects. Then, the distance between each sample box and the clustering center is calculated to assign the sample boxes. Finally, constant iterations are performed to update the clustering centers.

Since the K-means algorithm has a large randomness in the selection of the initial clustering centroids, this randomness will have a certain impact on the clustering results, leading to different results for each clustering, meaning the algorithm cannot determine the global optimal solution. Therefore, we used the K-means++ algorithm instead of K-means algorithm, and the selection of initial clustering centroids is improved to make the clustering better. The goal of the clustering algorithm is to make the intersection ratio (IOU) between the anchor box and the labeled box as large as possible, and the objective function uses IOU as the measure. The objective function is shown in Equation (7).
(7)$$f=\min\sum_{p=1}^{k}\sum_{q=1}^{n}\left[1-\mathrm{IOU}\right]$$

In Equation (7), p is the cluster center, k is the number of cluster centroids, q is the target frame of the sample labels, and n is the number of samples in the dataset. The relationship between the number of clustering centers and the average intersection ratio of the K-means++ algorithm on the dataset of this paper is shown in Figure 8a. The horizontal coordinate is the number of anchor boxes K, and the vertical coordinate is the value of the average intersection ratio IOU.

We found that as the value of K increases, the value of IOU also increases. When K = 12, the value of the average cross-merge ratio IOU tends to level off, while too many values of K affect the speed of detection. Therefore, we finally determined the number of anchor boxes to be 12. The distribution of the dataset label boxes is shown in Figure 8b.

The values of the cluster center coordinates we obtained are the normalized data. Therefore, the cluster center coordinate values need to be multiplied by the size of the model input image to get the final anchor box. The input image size of the model in this paper is 416 × 416 pixels. Therefore, the clustering effect is shown in Figure 8c.

We rounded the obtained anchor box sizes and arranged them by size. Among them, the three sizes of anchor point boxes, 19 × 34, 19 × 59, and 31 × 49, correspond to the 104 × 104 target detection layer. The three sizes of anchor boxes, 40 × 59, 28 × 81, and 51 × 81, correspond to the 52 × 52 target detection layer. The anchor boxes of 31 × 118, 41 × 118, and 62 × 103 correspond to the 26 × 26 target detection layer. The anchor box sizes of 53 × 131, 72 × 140, and 92 × 122 correspond to the 13 × 13 target detection layer. The anchor box sizes are shown in Table 1.

### 2.4. Test Evaluation Index

In order to evaluate the performance of the ewe estrus crawling behavior recognition model proposed in this study, the following evaluation indexes were selected: precision, recall, average precision, mean average precision F1, FPS and model size, as shown in Equations (8)–(12).
(8)$$P=\frac{T_{P}}{T_{P}{+F}_{P}}\times 100\%$$
(9)$$R=\frac{T_{P}}{T_{P}{+F}_{N}}\times 100\%$$
(10)$$F1=2\times \frac{P\times R}{P+R}\times 100\%$$
(11)$$\mathrm{AP}=\int_{0}^{1}\left(P\times R\right)\mathrm{dR}$$
(12)$$\mathrm{mAP}=\frac{\sum_{1}^{n}\mathrm{AP}}{n}$$

In Equations (8)–(12), T_P_ represents the number of positive samples identified in the positive samples, F_P_ represents the number of positive samples identified in negative samples, F_N_ represents the number of positive samples identified as negative, n represents the number of categories, and P is precision, R is recall, F1 is the comprehensive evaluation index of precision and recall, AP is average precision, and mAP is the mean of average precision.

In the evaluation process of the network model, each category can draw the P–R curve according to P and R, and the average precision AP is the area between the P–R curve and the coordinate axis. The average value of AP of all categories is the value of mAP. Since category n in this paper is 1, AP is also equal to the mAP.

## 3. Results and Discussion

### 3.1. Test Platform and Parameter Setting

Our model is on the same computer during training, validation, and testing. The computer was configured with an operating system of Windows 10, a GPU of GeForceRTX2080, a processor of Inter(R) Core(TM) i7-9700k CPU@3.2 GHz, and a running memory of 32 G. Our model was built using the PyTorch deep learning framework (Facebook AI Research Institute, Menlo Park, California, USA), PyCharm development tool (JetBrains, Prague, Czech Republic), and implemented through the Python language. The computational framework uses CUDA version 10.0 (NVIDIA Corporation, Santa Clara, CA, USA).

The relevant parameters of the model are set as follows: the training sample batch size is 10, the initial learning rate is set to 0.001, the weight decay is set to 0.0005, the momentum decay is set to 0.9, the optimizer is SGD, and the total number of training iterations is 100.

### 3.2. Training of Multi-Target Detection Layer Network Model

Our model loss value variation curves during training are shown in Figure 9. From the figure, we can see that the initial loss value of the model reaches 1812.

The total loss value of the model during the training process is the superposition of the loss values of all target detection layers. Our proposed model not only has four target detection layers but also changes the residual structure, which causes the model to have such high loss values in the initial phase of training.

Although our model has a high initial loss value, the loss value decreases rapidly as the training proceeds. This is because the network model updates the node weights during the back-propagation process of training, and the updated weights make the model’s fitting ability gradually improve.

When the number of training iterations of the network model exceeds 80, the loss value gradually stabilizes between 2 and 2.5. The loss values of the subsequent model smooth out overall with only slight oscillations, indicating the completion of the network model training.

### 3.3. Determination of Optimal Model

When we train the model, a model weight is saved after one training iteration is completed. After the model training was completed, we obtained a total of 100 model weights. For the subsequent identification of ewe estrus climbing behavior, we need to evaluate the obtained models using evaluation metrics to find the optimal detection model. Figure 10 shows the P, R, F1, and mAP of the model at each training.

We can see from the figure that the overall trend of the model is growing during the first 60 training iterations, although the variation of each metric is large. After 80 training iterations, the overall metrics of the models gradually stabilized. The mAP of several of the models reached above 99%. 

After considering the F1 value and other three indexes, we finally chose the model saved after 96 training sessions as the ewe estrus climbing behavior recognition model in this paper.

### 3.4. Comparative Analysis of Different Confidence Threshold

We also need to compare the values of P, R, F1, and AP of this model under different confidence thresholds after selecting the optimal model.

From Figure 11a, we can see that the precision increases with the confidence threshold and reaches a maximum at a threshold value of 1. From Figure 11b, we see that the recall decreases as the confidence threshold increases and reaches a minimum at a threshold value of 1.

When the confidence threshold is set larger, our model will exhibit higher precision, lower recall, and higher ewe estrus misidentification. When the confidence threshold is set smaller, our model will exhibit lower precision, higher recall, and lower ewe estrus misidentification.

This phenomenon leads us to set the confidence threshold of the model as neither too large nor too small.

Therefore, we mainly refer to the F1 value of the combined evaluation index of precision and recall in the selection of confidence threshold.

From Figure 11c, we can see that F1 has a high value when the confidence threshold of the model is 0.5. In summary, we determined the optimal confidence threshold of the model to be 0.5. At this point, the P of the model was 98.56%, the R was 98.04%, and the F1 value was 98%.

Figure 11d shows the model AP value of 99.78% at a confidence threshold of 0.5, and the mAP value of 99.78% since the dataset category was 1.

### 3.5. Comparative Analysis of Different Models

To analyze the variability of our model with other models, we used faster R-CNN (Resnet50), faster R-CNN (VGG16), YOLO v3 and YOLO v4 to compare the performance with our model. In the comparative analysis, we used P, R, F1, mAP, FPS, and model size as evaluation metrics, which were validated on the same test set. The results are shown in Table 2.

We can see the advantages and disadvantages of our model with other models from the data in the Table 2. Compared with the faster R-CNN (Resnet50) model, our proposed model has a 4% increase in F1, a small difference in mAP, a 9.41% increase in P, a 1.78% decrease in R, and a 36 f/s increase in FPS, with model size increasing by 168 MB. Compared with the faster R-CNN (VGG16), our model F1 was increased by 9%, mAP is not much different, P was increased by 18.36%, R was decreased by 1.78%, FPS was increased by 29 f/s, and model size was reduced 245 MB.

Compared with YOLO v3, our model F1 was increased by 1%, the mAP is not much different, the P was increased by 3.7%, the R was decreased by 0.89%, the FPS was decreased by 4 f/s, and the model size was increased by 41 MB. Compared with YOLO V4, our proposed model F1 was increased by 2%, mAP is not much different, P was increased by 0.44%, R was increased by 4.83%, FPS was increased by 4.77 f/s, and model size was increased by 32 MB.

Based on the original YOLO v3 model, our model adds a 104 × 104 target detection layer to detect distant ewes in the video image, enhancing the model’s ability to detect small targets. At the same time, we increase the number of cells in the residual structure of the model. On the one hand, it prevents the loss of shallow feature information that occurs during the forward-propagation training of the model. On the other hand, we prevent the problem of gradient dispersion and gradient explosion during the backward-propagation training of the model. The fitting capability of the model is enhanced.

In summary, our model can obtain more deep and shallow feature information, and the deep and shallow feature data are utilized with each other, which increases the learning ability and performance of the model. It enables our model to improve both P and F1 values for ewe estrus climbing across behavior recognition, and the detection speed FPS was 41 f/s, which can meet the requirements of real-time recognition.

### 3.6. Analysis of Identification Results

In order to further validate the performance of the model in the recognition of estrus climbing behavior of ewes, we perform a comparative analysis in terms of false recognition, occlusion, and long distances.

Figure 12 shows the misidentification results of the model. We can see from Figure 12b,c that there is misidentification in the YOLO v3 and YOLO v4 model, which detect ewes that are not in heat as being in heat. Figure 12d shows our model recognition results. Our model was able to identify ewes in estrus located at the edge of the video image, with 67% estrus detection for ewes and no false identification.

In conclusion, our proposed model has lower false identification results than the original model in detecting ewes in estrus and can correctly identify the climbing behavior of ewes in estrus.

Figure 13 shows the recognition results of the model in the presence of occlusion.

We can see from Figure 13 that YOLO v3, YOLOv4 and our model are able to identify ewes in estrus in the video image under the condition of slight occlusion by other ewes, with 85%, 100% and 100% identification of ewes in estrus climbing across behavior. The recognition results in Figure 13b,d show that our proposed model improves the recognition rate by 15% compared to that shown by YOLO v3. This is because we optimized the number of residual structures and improved the fusion of deep and shallow feature information.

From the recognition results in Figure 13c,d we can see that both YOLO v4 and our model recognize 100% of ewes in estrus climbing behavior, but our model is faster in detecting ewes in estrus.

Figure 14 shows the recognition results of the model in the presence of long distances.

We can see from Figure 14 that YOLO v3, YOLO v4 and our model are able to identify ewes in estrus in the video images when the ewe estrus climbing behavior occurs at a longer distance. However, YOLO v3 and YOLO v4 showed 85% and 95% detection of ewe estrus climbing behavior, while our proposed model showed 100% detection of ewe estrus climbing behavior, which is due to our model’s increased target detection layer and improved ability to detect distant estrus ewes.

We can see the advantages of our model from the recognition results, which are mainly due to our model’s additional layer of target detection and increased number of residual units compared to the original model.

We know that the shallow feature layer is rich in detail information such as color, texture, edge and target location, while the deep feature layer has complex and abstract semantic information.

Our improved method allows the model to further integrate shallow and deep feature information, thus improving the model’s ability to identify estrus crawling behavior in distant ewes.

## 4. Conclusions

The timely and accurate detection of estrus behavior in ewes is an important research topic in precision animal husbandry. Our research work has provided some assistance in the detection of ewe estrus crawling behavior in large-scale meat sheep farming and has led to the following conclusions:

(1) We have made some improvements based on the YOLO v3 model structure. We improve the ability of the model to detect distant estrus ewes in video images by increasing the target detection layer of the model. By increasing the number of cells in the residual structure of the model, we improve the model’s ability to fuse shallow feature information with deep feature information. We reduce the impact of image distortion on the model by performing an image resolution maintenance aspect ratio operation on the model data-loading module. We made some improvements to the model’s structure through the abovementioned operation to improve the ability of the model to detect estrus crawling behavior of ewes.

(2) We re-clustered the anchor box size of the model to further improve its performance, and finally obtained the ewe estrus crawling behavior detection model. Our model shows better performance in comparison tests with models such as faster R-CNN (Resnet50), faster R-CNN (VGG16), YOLO v3 and YOLO v4.

In conclusion, we very much hope that our proposed model/method can provide some help and reference for estrus behavior detection of ewes in large-scale meat sheep farming.

## Figures and Tables

**Figure 1 animals-13-00413-f001:**
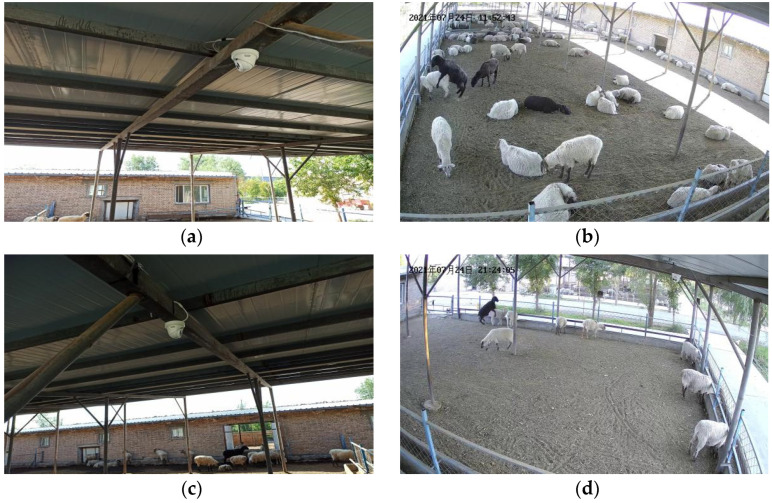
Camera installation position and field of view image: (**a**) Camera 1 installation position; (**b**) Camera 1 view image; (**c**) Camera 2 installation position; (**d**) Camera 2 view image. The Chinese characters in the upper left corner of the image are the date of the video recording, in order of year, month, and day.

**Figure 2 animals-13-00413-f002:**
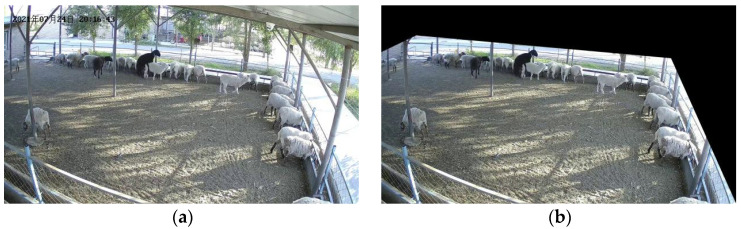
Video image mask operation: (**a**) Image before mask; (**b**) Image after mask. The Chinese characters in the upper left corner of the image are the date of the video recording, in order of year, month, and day.

**Figure 3 animals-13-00413-f003:**
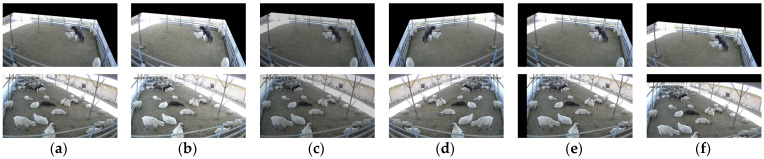
Data enhancement processing: (**a**) Original picture; (**b**) Brightness enhancement; (**c**) Brightness reduction; (**d**) Flip horizontally; (**e**) Horizontal movement; (**f**) Vertical movement. The Chinese characters in the upper left corner of the image are the date of the video recording, in order of year, month, and day.

**Figure 4 animals-13-00413-f004:**
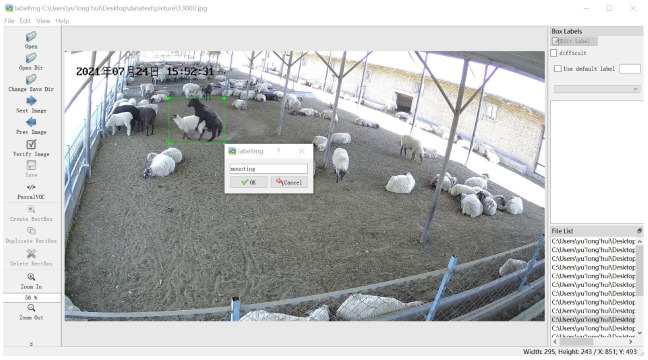
Dataset label making. The Chinese characters in the upper left corner of the image are the date of the video recording, in order of year, month, and day.

**Figure 5 animals-13-00413-f005:**
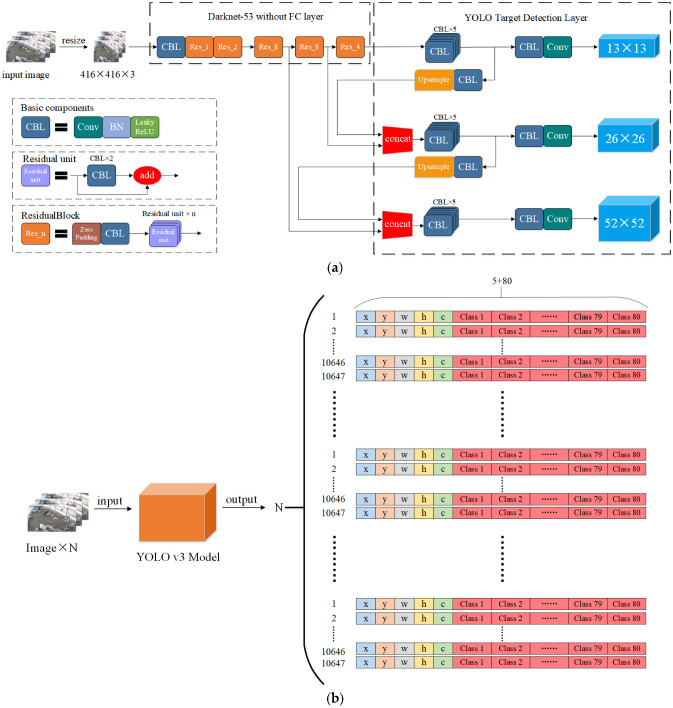
Structure and output of YOLO v3 model: (**a**) Structure of YOLO v3 model; (**b**) Output of YOLO v3 model.

**Figure 6 animals-13-00413-f006:**
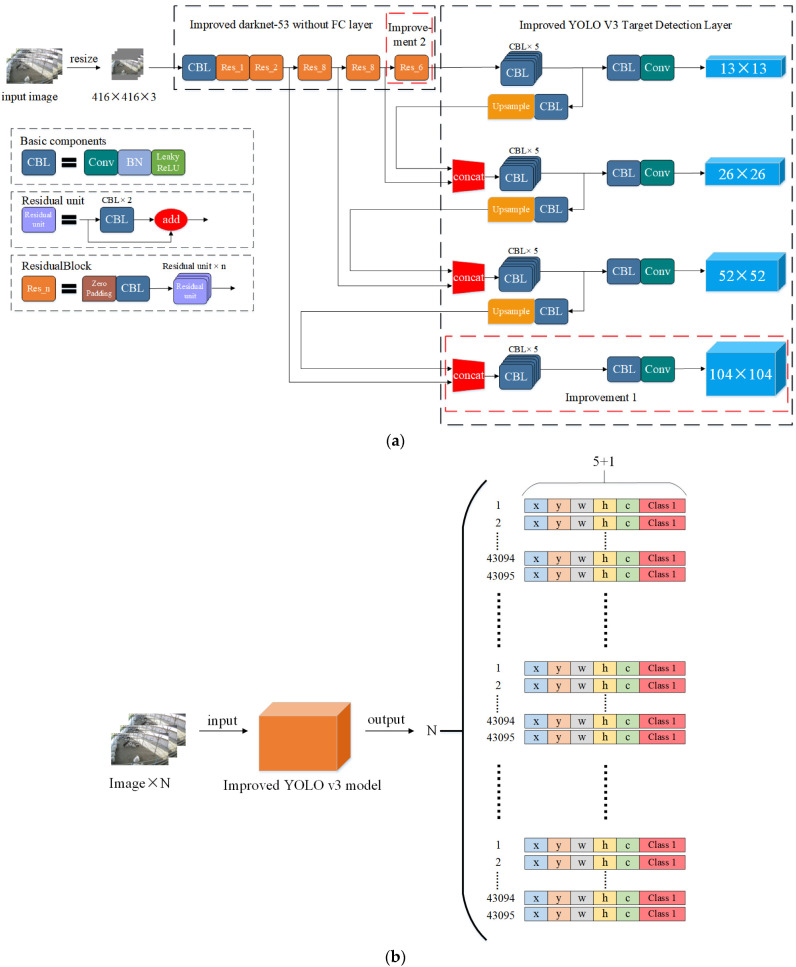
Structure and output of multi-target detection layer network model: (**a**) Structure of multi-target detection layer network model; (**b**) Output of multi-target detection layer network model.

**Figure 7 animals-13-00413-f007:**
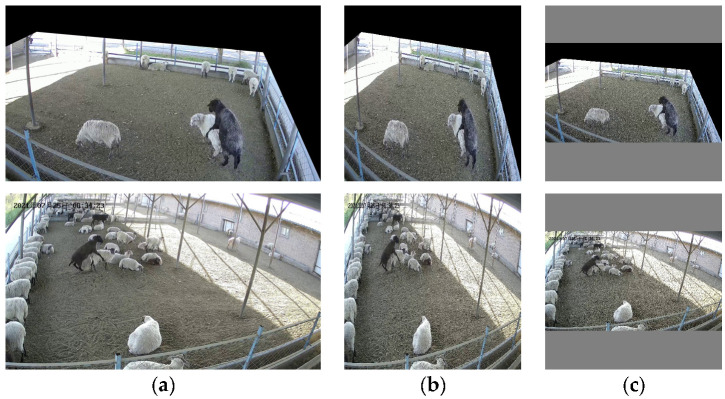
Picture resolution adjustment: (**a**) Original picture; (**b**) Direct adjustment; (**c**) Maintaining adjustment of aspect ratio.

**Figure 8 animals-13-00413-f008:**
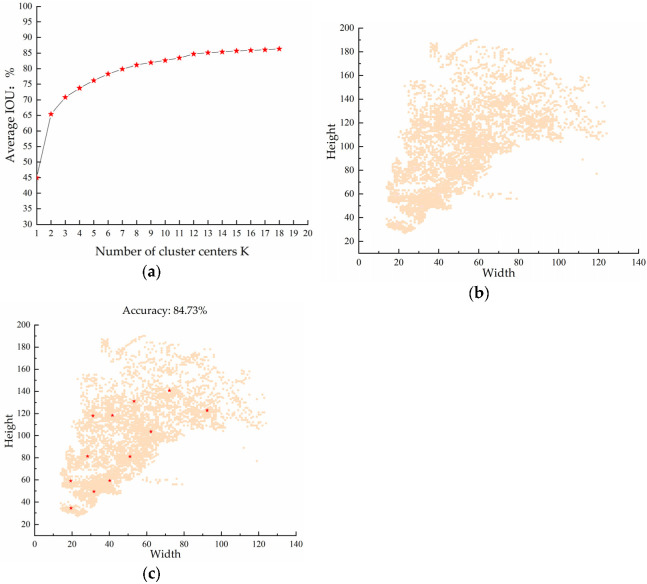
Anchor box re-clustering of datasets: (**a**) The average IOU values corresponding to different K values; (**b**) Distribution of labeled boxes of the dataset; (**c**) Clustering effect of anchor boxes.

**Figure 9 animals-13-00413-f009:**
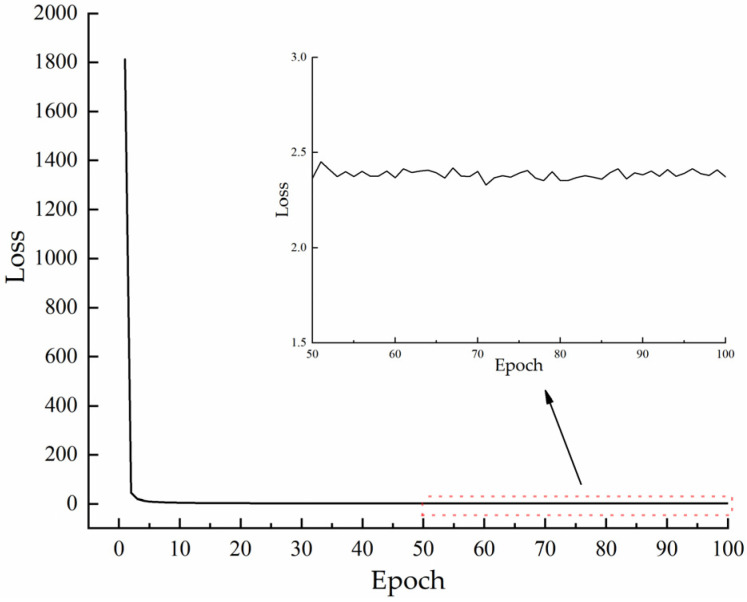
Variation of model loss values with the number of training iterations.

**Figure 10 animals-13-00413-f010:**
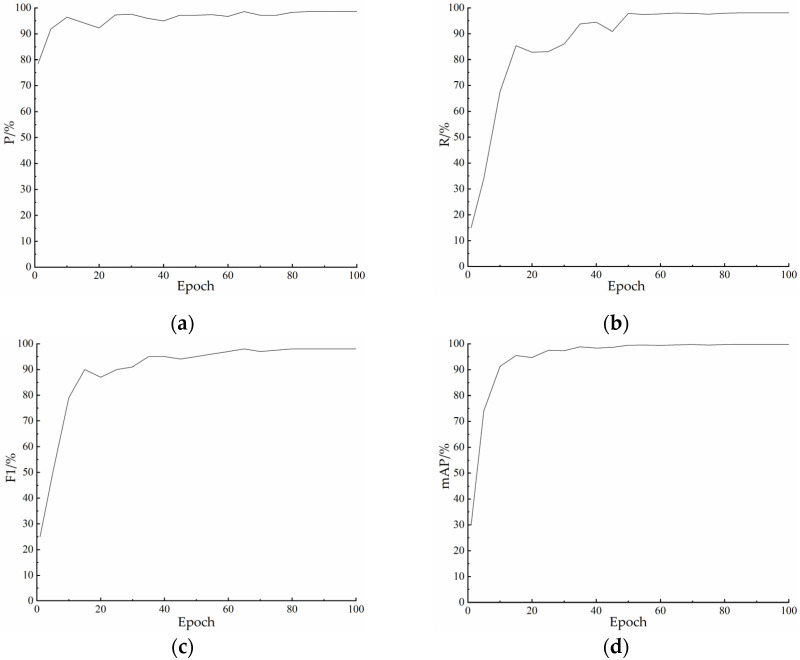
Variation of each index with training times: (**a**) Curve of P value changing with training times; (**b**) Curve of R value changing with training times; (**c**) Curve of F1 value changing with training times; (**d**) Curve of mAP value changing with training times.

**Figure 11 animals-13-00413-f011:**
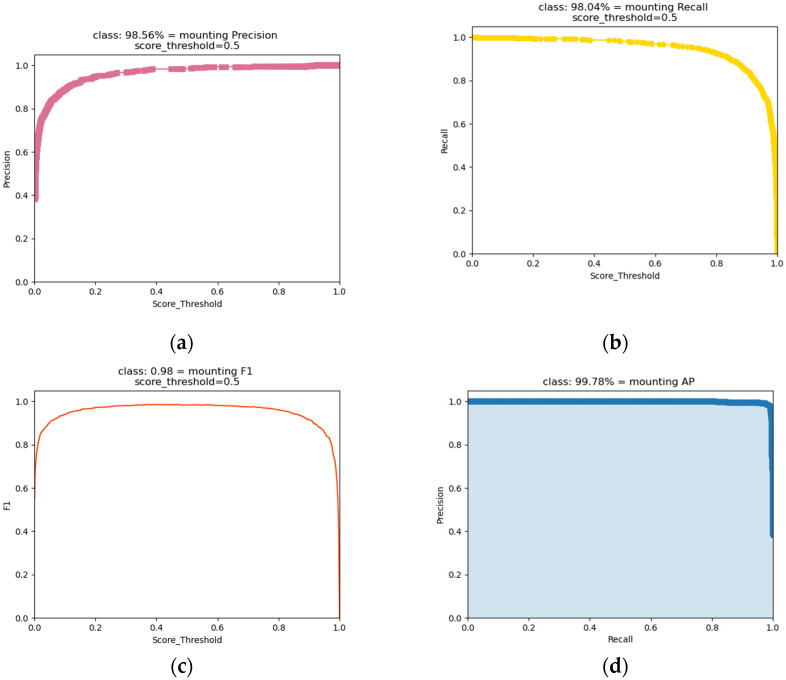
Variation of each index with threshold: (**a**) Curve of P value changing with threshold; (**b**) Curve of R value changing with threshold; (**c**) Curve of F1 value changing with threshold; (**d**) Curve of AP value changing with threshold.

**Figure 12 animals-13-00413-f012:**
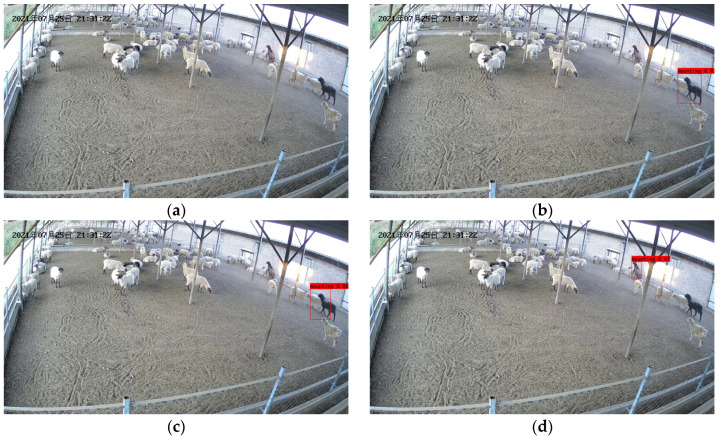
False identification results of the model: (**a**) Original image; (**b**) YOLO v3 recognition results; (**c**) YOLO v4 recognition results; (**d**) Identification results of our model. The Chinese characters in the upper left corner of the image are the date of the video recording, in order of year, month, and day.

**Figure 13 animals-13-00413-f013:**
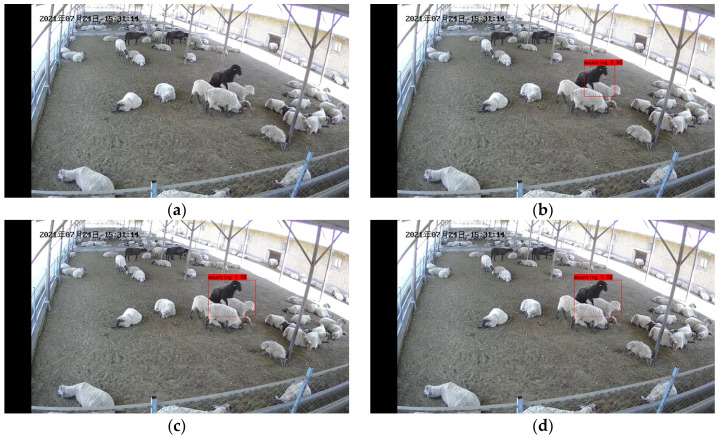
Recognition results in the presence of occlusion: (**a**) Original image; (**b**) YOLOv3 recognition results; (**c**) YOLO v4 recognition results; (**d**) Identification results of our model. The Chinese characters in the upper left corner of the image are the date of the video recording, in order of year, month, and day.

**Figure 14 animals-13-00413-f014:**
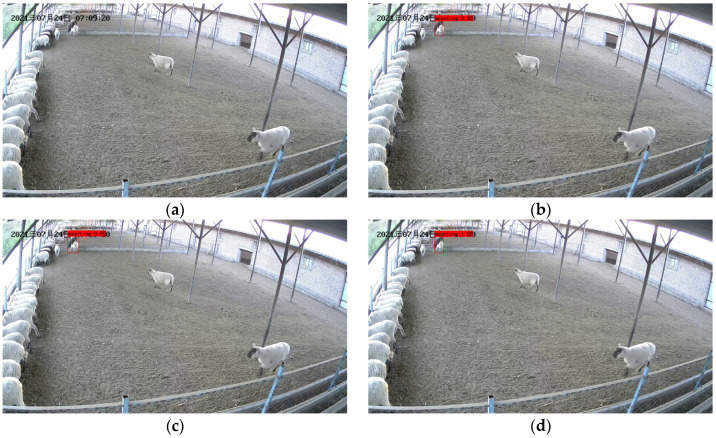
Recognition results in the presence of long distances: (**a**) Original image; (**b**) YOLOv3 recognition results; (**c**) YOLO v4 recognition results; (**d**) Identification results of our model. The Chinese characters in the upper left corner of the image are the date of the video recording, in order of year, month, and day.

**Table 1 animals-13-00413-t001:** Clustering center coordinate values and anchor box size values.

Cluster Center	Clustering Center Coordinate Value	Anchor Box Size	Target Detection Layer
1	(0.045313, 0.081250)	19 × 34	
2	(0.046094, 0.140972)	19 × 59	104 × 104
3	(0.074219, 0.118056)	31 × 49	
4	(0.095313, 0.140972)	40 × 59	
5	(0.068359, 0.195139)	28 × 81	52 × 52
6	(0.122656, 0.194444)	51 × 81	
7	(0.074414, 0.282639)	31 × 118	
8	(0.098437, 0.283333)	41 × 118	26 × 26
9	(0.148437, 0.247222)	62 × 103	
10	(0.127734, 0.314583)	53 × 131	
11	(0.173828, 0.336805)	72 × 140	13 × 13
12	(0.221094, 0.294444)	92 × 122	

**Table 2 animals-13-00413-t002:** Comparison of evaluation indexes of this model and other models.

Evaluation Index	P	R	F1	mAP	FPS	Model Size
Faster R-CNN (Resnet50)	89.15%	99.82%	94%	99.48%	5 f/s	108 MB
Faster R-CNN (VGG16)	80.20%	99.82%	89%	99.41%	12 f/s	521 MB
YOLO v3	94.86%	98.93%	97%	99.29%	45 f/s	235 MB
YOLO v4	98.12%	93.21%	96%	99.14%	36.23 f/s	244 MB
Our Model	98.56%	98.04%	98%	99.78%	41 f/s	276 MB

## Data Availability

The data presented in this study are available on request from the corresponding author. The data are not publicly available due to being part of an ongoing study.

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
