# Peer review of "A Recognition Method of Ewe Estrus Crawling Behavior Based on Multi-Target Detection Layer Neural Network"

_animals, 2023, doi:10.3390/ani13030413_

Round 1

Reviewer 1 Report

Suggestions for modification are as follows:

1.     Throughout the paper, you have studied the method of identifying the ewe's crawling behavior, and thus determining its estrus behavior. It is suggested to change the title to: A recognition method of ewe estrus crawling behavior based on multi-target detection layer neural network.  The expression of relevant descriptions in the text and conclusion should be modified. Because the recognition method of estrus behavior is not only a recognition method of crawling behavior.

2.     The article does not mention why YOLOv3 was selected for optimization, so it is suggested to add.

3.     It is suggested that Figure 8 be improved to show the effect before and after clustering.

4.     In line 338, the total number of training iterations is 100.  Why set the number of training number is 100, is it the empirical value after repeated training, or the theoretical optimal value?

5.     The resolution of the proposed image maintains the aspect ratio, similar to that of YOLOv5. In addition, the YOLOv5 model is smaller and easier to deploy. How do you consider this?

6.     As can be seen from Table 2, YOLO v4 is not much different from your model in terms of P, F1, mAP, FPS and model size. In the analysis of model identification results, it is suggested to add the identification comparison and analysis of YOLO v4, YOLO v3 and your model.

Author Response

Dear Reviewer:

Thanks very much for taking your time to review our manuscript entitled “A recognition method of ewe estrus crawling behavior based on multi-target detection layer neural network” (ID: animals-2067367).

We appreciate all your comments and suggestions. Those comments are all valuable and very helpful for revising and improving our paper, as well as the important guiding significance to our research. We have studied the comments carefully and have made the corrections which we hope meet with approval. We have responded to your suggestions as follows:

Point 1: Throughout the paper, you have studied the method of identifying the ewe's crawling behavior, and thus determining its estrus behavior. It is suggested to change the title to: A recognition method of ewe estrus crawling behavior based on multi-target detection layer neural network. The expression of relevant descriptions in the text and conclusion should be modified. Because the recognition method of estrus behavior is not only a recognition method of crawling behavior.

Response 1: We are very grateful to you for this suggestion. We have not considered the title of the paper fully, and your proposed method of identifying estrus behavior in ewes is correct not only for climbing across behavior. We are inspired by your suggestion and will try other methods of ewe estrus detection in our future studies. Once again, we would like to express our sincere thanks to you.

We have revised the title of the paper based on your suggestions, and also revised the relevant expressions in the text, which can be seen in the paper in the 2, 31, 109, 112, 121, 132, 135, 154, 548, 557, 560, and so on line.

Point 2: The article does not mention why YOLOv3 was selected for optimization, so it is suggested to add.

Response 2: We thank you very much for this suggestion. We have added the relevant information based on your suggestion, which can be seen in the 98-106 line in the paper.

Point 3: It is suggested that Figure 8 be improved to show the effect before and after clustering.

Response 3: We thank you very much for this suggestion. We have made changes based on your suggestions, which can be seen in the 330-332 line in the paper.

Point 4: In line 338, the total number of training iterations is 100.  Why set the number of training number is 100, is it the empirical value after repeated training, or the theoretical optimal value?

Response 4: We thank you very much for this question. We are here to answer you why the training count is set to 100. We train the model as we modify it, and after many training sessions, we find that the model training count of 100 times is achievable.

Point 5: The resolution of the proposed image maintains the aspect ratio, similar to that of YOLOv5. In addition, the YOLOv5 model is smaller and easier to deploy. How do you consider this?

Response 5: We thank you very much for this question. We provide you with a brief answer here: 1. We think YOLO v3 is a very classic network, which is more stable, more widely used in engineering applications, and more mature in engineering application technology. 2. We have some knowledge of YOLO v5, which is a relatively new model and a relatively small model, but our code implementation of it is not yet mature. Our answers may not be very detailed, and we sincerely apologize to you.

Thank you very much for this question, which has inspired us. Inspiration 1: We try to use models like YOLO v5 for estrus detection in ewes in our future research work. Inspiration 2: We will try to prune the model, reduce the parameters of the model, and perform engineering deployment. Once again, we would like to express our sincere gratitude to you.

Point 6: As can be seen from Table 2, YOLO v4 is not much different from your model in terms of P, F1, mAP, FPS and model size. In the analysis of model identification results, it is suggested to add the identification comparison and analysis of YOLO v4, YOLO v3 and your model.

Response 6: We are very grateful to you for this suggestion. We have made changes based on your suggestion, which can be seen in line 483-542 in the paper.

We once again express our heartfelt thanks to you.

Reviewer 2 Report

General Comments

The study reports a recognition method of ewe estrus behavior using image. The method can be used to estrus behavior detection in some specific conditions of sheep breeding. Additionally, provides support for develop different strategies on other studies. Although the detailed description of the steps used by the authors on Material and Methods section, I suggest adding information about the dataset (e.g., the number of images with distant ewes in estrus, close ewes in estrus and/or outside the detection frame on training, validation, and testing). This description can be used on discussion to support the difference observed between the YOLO v3 and proposed method, as the difference was not so high. Also, the authors compared the results with different methods, but the discussion is only compared to the YOLO v3 method. I understood that the proposed method is YOLO v3 modified, then the authors focused the comparison on that model. However, the discussion with the other methods used should enrich the paper.

Specific comments

Lines 63-64: “…properties of sensors [9-19]. movement…” Please, check the sentence.

Lines 98-99: “… ewe estrus detection, we proposed a multi-objective…”.

Line 155: “…three frames to obtain 3640 original images.”.

Line 157: Did these other behaviors are from the other animals that were not involved to estrus behavior on that moment? If so, I think this information should be added on the sentence.

Line 172: “…of dataset labels…”.

Line 178: “…of 7:2:1 [38], there were…”.

Line 180: “dataset labels were created…”.

Line 220: The first sentence doesn’t seem connected. Please, check what you want to inform with this sentence.

Equation 2: Although N and M are described on Equation 1, I suggest to authors describe after Equation 2.

Line 262: “… as shown between the Equations (3) and (6):”.

Line 263: “In Equations (3) to (6)…”.

Line 269: “…image high. When…”.

Lines 287-289: “Due to the target size in the COCO dataset does not match with the target size in this dataset, we needed to …”

Line 344-351: I suggest checking the connectedness among sentence in this paragraph. Although it is possible to understand by the context, it is confusing.

Line 366: “…in Equations (8) to (12)”.

Equation 10: Please, check the numerator of division. I think that there is a symbol missing, which would be a multiplication, right?

Lines 361-373: In my opinion, this should be on Materials and Methods section, once the authors describe the metrics (methods) used to evaluate the model.

Figure 11: The word “therehold” should be corrected. Also, the authors don’t show the threshold changing on Figure 11d as mentioned on description, once the x axis is labeled as Recall and y axis as Precision.

Lines 396 and 406: Do you mean precision when use the “accuracy” word? If so, please keep the name of metric (precision) to avoid misunderstanding.

Line 422: “…data in the Table 2.”.

Line 452: Do you mean false identification result for YOLO3 on the recognition at the upper right of the image? If so, please add this information on the sentence.

Figure 12: If I understood correctly, the false identification results are only for YOLOv3. Therefore, I suggest rewritten the figure description evidencing this point.

Author Response

Dear Reviewer:

Thanks very much for taking your time to review our manuscript entitled “A recognition method of ewe estrus crawling behavior based on multi-target detection layer neural network” (ID: animals-2067367).

We appreciate all your comments and suggestions. Those comments are all valuable and very helpful for revising and improving our paper, as well as the important guiding significance to our research. We have studied the comments carefully and have made the corrections which we hope meet with approval. We have responded to your suggestions as follows:

Point 1: The study reports a recognition method of ewe estrus behavior using image. The method can be used to estrus behavior detection in some specific conditions of sheep breeding. Additionally, provides support for develop different strategies on other studies. Although the detailed description of the steps used by the authors on Material and Methods section, I suggest adding information about the dataset (e.g., the number of images with distant ewes in estrus, close ewes in estrus and/or outside the detection frame on training, validation, and testing). This description can be used on discussion to support the difference observed between the YOLO v3 and proposed method, as the difference was not so high. Also, the authors compared the results with different methods, but the discussion is only compared to the YOLO v3 method. I understood that the proposed method is YOLO v3 modified, then the authors focused the comparison on that model. However, the discussion with the other methods used should enrich the paper.

Response 1: We are very grateful for your suggestion. We have given careful thought to your suggestions.

Here we answer:

  1. The suggestion to supplement the information of the dataset, we did not consider comprehensively when making the dataset at that time, this suggestion of yours is very good, we will deeply analyze the dataset and supplement the missing information in our future work.
  2. Add the discussion of other identification methods. We have added the discussion of other methods according to your suggestion, which can be seen in the text 483-542 line.

We apologize that our answer may not be very comprehensive. We thank you again for your suggestions.

Point 2: Lines 63-64: “…properties of sensors [9-19]. movement…” Please, check the sentence.

Response 2: We are very grateful for your suggestion. We changed "." to "," based on your suggestion, and marked in the paper 64 line.

Point 3: Lines 98-99: “ewe estrus detection, we proposed a multi-objective”.

Response 3: We are very grateful to you for this suggestion. We have followed your suggestion to change "." to "," and lowercase the first letter. These changes are visible in the text 108 line.

Point 4: Line 155: “…three frames to obtain 3640 original images.”.

Response 4: We are very grateful for your suggestion. We have removed the "," based on your suggestion, and the change is visible in the text 164 line.

Point 5: Line 157: Did these other behaviors are from the other animals that were not involved to estrus behavior on that moment? If so, I think this information should be added on the sentence.

Response 5: We really appreciate you asking this question. These behaviors are those of other ewes. The other ewes were not in estrus at the time. We have added the relevant information to the sentence based on your suggestion, which is visible in the text 166 line.

Point 6: Line 172: “…of dataset labels…”.

Response 6: We thank you very much for this suggestion. We have removed the spaces according to your suggestion, which can be seen in line 182 in the paper.

Point 7: Line 178: “…of 7:2:1 [38], there were…”.

Response 7: We are grateful for your suggestion. We have changed "are" to "were" based on your suggestion, which can be seen in line 188 of the paper.

Point 8: Line 180: “dataset labels were created…”.

Response 8: We are grateful for your suggestion. We have changed "are" to "were" based on your suggestion, which can be seen in line 190 of the paper.

Point 9: Line 220: The first sentence doesn’t seem connected. Please, check what you want to inform with this sentence.

Response 9: We thank you very much for this suggestion. We have removed the first sentence based on your suggestion, which can be seen in line 231 in the paper.

Point 10: Equation 2: Although N and M are described on Equation 1, I suggest to authors describe after Equation 2.

Response 10: We are very grateful to you for this suggestion. We have placed the descriptions of N and M after Equation 2 as per your suggestion, which can be seen in the 243-244 line in the paper.

Point 11: Line 262: “… as shown between the Equations (3) and (6):”.

Response 11: We are very grateful for your suggestions. We have made changes based on your suggestions, which can be seen in line 275 of the paper.

Point 12: Line 263: “In Equations (3) to (6)…”.

Response 12: We thank you very much for this suggestion. We have changed "and" to "to" according to your suggestion, which can be seen in line 276 of the paper.

Point 13: Line 269: “…image high. When…”.

Response 13: We thank you very much for this suggestion. We have changed "," to "." according to your suggestion, which can be seen in line 282 of the paper.

Point 14: Lines 287-289: “Due to the target size in the COCO dataset does not match with the target size in this dataset, we needed to …”

Response 14: We would like to thank you not only for this suggestion, but also for revising the statement for us. We have made changes based on your suggestions, which can be seen in line 300-302 of the paper. Once again, we would like to express our sincere thanks to you.

Point 15: Line 344-351: I suggest checking the connectedness among sentence in this paragraph. Although it is possible to understand by the context, it is confusing.

Response 15: We are very grateful for your suggestions. We have made changes based on your suggestions, which can be seen in line 381-384 of the paper.

Point 16: Line 366: “…in Equations (8) to (12)”.

Response 16: We thank you very much for this suggestion. We have changed "and" to "to" according to your suggestion, which can be seen in line 346 of the paper.

Point 17: Equation 10: Please, check the numerator of division. I think that there is a symbol missing, which would be a multiplication, right?

Response 17: We are very grateful for your suggestion. We checked the molecules and found that a multiplication sign was indeed missing. Furthermore, we have made the change based on your suggestion, which can be seen in line 346-347 in the paper.

Point 18: Lines 361-373: In my opinion, this should be on Materials and Methods section, once the authors describe the metrics (methods) used to evaluate the model.

Response 18: We are very grateful for your suggestions. We have made changes based on your suggestions, which can be seen in line 342-356 of the paper.

Point 19: Figure 11: The word “therehold” should be corrected. Also, the authors don’t show the threshold changing on Figure 11d as mentioned on description, once the x axis is labeled as Recall and y axis as Precision.

Response 19: We thank you very much for this suggestion. We have changed the word "therehold" to "threshold" in the figure as you suggested, which can be seen in line 435-436 of the paper.

We are very grateful for your careful review. In Figure 11d, there is an irregularity in our description, and we express our sincere apologies to you. What we mainly want to express is that in the graph the X-axis is the recall R and the Y-axis is the precision P. The variation of the AP value of the area shaded by the dark blue line is influenced by P and R, while the values of P and R are influenced by the threshold value.

We may not have expressed the content in great detail. Again, we sincerely apologize to you, and thank you again for your careful review.

Point 20: Lines 396 and 406: Do you mean precision when use the “accuracy” word? If so, please keep the name of metric (precision) to avoid misunderstanding.

Response 20: We are very grateful for your suggestions. We have made changes based on your suggestions, which can be seen in line 425-432 of the paper.

Point 21: Line 422: “…data in the Table 2.”.

Response 21: We are very grateful for your suggestions. We have made changes based on your suggestions, which can be seen in line 455 of the paper.

Point 22: Line 452: Do you mean false identification result for YOLO3 on the recognition at the upper right of the image? If so, please add this information on the sentence.

Response 22: We are very grateful to you for this suggestion. We have made changes based on your suggestion, which can be seen in line 483-487 in the paper.

Point 23: Figure 12: If I understood correctly, the false identification results are only for YOLOv3. Therefore, I suggest rewritten the figure description evidencing this point.

Response 23: We are very grateful to you for this suggestion. We have made changes based on your suggestion, which can be seen in line 483-497 in the paper.

We once again express our heartfelt thanks to you.